# A Systematic Evaluation of Preference Aggregation in Federated RLHF for Pluralistic Alignment of LLMs

## Abstract

This paper addresses the challenge of aligning Large Language Models (LLMs) with diverse human preference within Federated Learning (FL) environments, where standard methods often fail to adequately represent diverse viewpoints. We introduce a comprehensive evaluation framework that systematically assesses the trade-off between alignment quality and fairness when using different aggregation strategies for human preferences. Specifically, we evaluate standard aggregation techniques Min, Max, and Average 0and introduce a novel adaptive scheme that dynamically adjusts preference weights based on a group's historical alignment performance. Our experiments on Q/A tasks using a Proximal Policy Optimization (PPO)-based RLHF pipeline demonstrate that our adaptive approach consistently achieves superior fairness, while maintaining competitive alignment scores. This work offers a robust methodology for evaluating LLM behavior across diverse populations and provides a practical solution for developing truly pluralistic and fairly aligned models.

## 1 Introduction

The remarkable capabilities of LLMs have positioned them as a central technology across various domains. However, their real-world utility and safety hinge on their ability to align with complex and diverse human values and social norms Yang et al. (2024); Sorensen et al. (2024). The prevailing methodology for this alignment is Reinforcement Learning from Human Feedback (RLHF), which fine-tunes models based on collected human preference data Ouyang et al. (2022). While effective, the standard RLHF paradigm often operates on a centralized dataset, which is not only a privacy concern but also risks embedding biases of a narrow demographic Casper et al. (2023).

To address this, the integration of RLHF with Federated Learning (FL) has emerged as a promising avenue. FL allows for model training on decentralized data from numerous clients, thus preserving data privacy and capturing a wider range of human preferences Wu et al. (2024); Srewa et al. (2025). However, this fusion presents a critical and underexplored challenge: **How to aggregate the diverse and potentially conflicting preference signals from different user groups?** The choice of aggregation strategy is not merely a technical detail; it is an evaluation protocol that directly shapes the model's final behavior, determining whose preferences are prioritized and whose are marginalized.

This paper proposes a systematic evaluation framework to analyze the impact of different aggregation techniques on both alignment performance and fairness. By comparing standard methods with our proposed adaptive aggregation scheme, our goal is to define a more robust protocol to assess LLMs in decentralized, pluralistic environments. We show that while simple aggregation methods can lead to unintended biases, our adaptive approach strikes a superior balance between achieving strong overall alignment and ensuring equitable representation across diverse groups, thus contributing to the development of more reliable and justly aligned LLMs. Our approach follows a zero-shot alignment paradigm, using only aggregated group reward signals without task demonstrations, ensuring generalizable alignment.

## 2 Background and Related Work

The alignment of LLMs with complex human preferences is a central goal in their development. Since explicitly defining human values in a loss function is challenging, a robust paradigm has emerged in which models learn directly from human preference data. RLHF has become a powerful technique for this purpose, guiding LLMs toward desired behaviors like safety and helpfulness Ouyang et al.

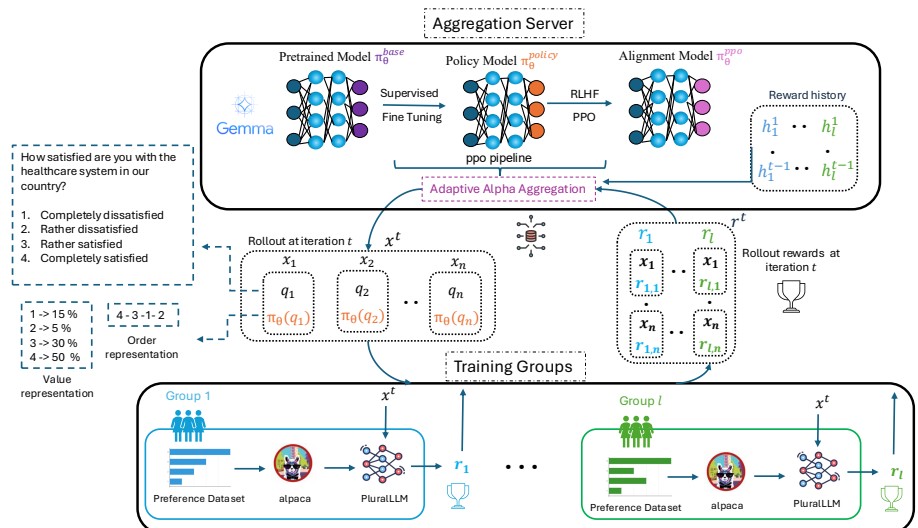

Figure 1: Federated RLHF for pluralistic alignment of group preferences in LLM.

(2022). The most common RL algorithm used in RLHF is PPO which fine-tunes the LLM by receiving feedback from a reward model trained on human preference pairs Schulman et al. (2017). An alternative, Direct Preference Optimization (DPO), simplifies this process by bypassing the need for a separate reward model, instead directly optimizing the model to assign a higher probability to preferred responses Rafailov et al. (2023).

Addressing the issue of diverse preferences and potential underrepresentation, methods like Group Robust Policy Optimization (GRPO) Ramesh et al. (2024) and MaxMin-RLHF Chakraborty et al. (2024) have been introduced to ensure robust alignment across various user groups. However, these methods, like the traditional RLHF pipeline, are typically centralized, which poses significant privacy risks by requiring the collection of user data on a central server. This privacy concern has spurred the development of decentralized approaches. PluralLLM Srewa et al. (2025), for instance, uses federated learning to train a transformer-based preference predictor, allowing different user groups to collaboratively align a model without sharing their sensitive raw preference data. Our work builds upon this foundation by moving beyond simple aggregation to systematically evaluate the impact of different aggregation techniques on both alignment performance and fairness within a federated RLHF framework. Further implementation details of the preference predictor are provided in Appendix B

## 3 METHODOLOGY

**System Setup and Training Groups:** Our framework focuses on Q/A tasks with $l$ training groups $G_{\text{train}} = \{g_1, g_2, \ldots, g_l\}$, where each group $g_i$ maintains its private preference dataset $D_{g_i} = \{(x_j, y_j)\}$ locally. Each preference sample consists of a query-response pair embedding $x_j$ and the corresponding group preference probability $y_j$. These datasets are distributed across groups and never shared with the central server, ensuring privacy preservation.

As illustrated in Figure 1, the aggregation server is initialized with a base LLM model $\pi_\theta^{base}$ and performs supervised fine-tuning (SFT) to adapt it for Q/A tasks, resulting in a policy model $\pi_\theta^{policy}$ suitable for PPO training. The server coordinates between policy optimization and distributed preference learning.

At iteration $t$, the server generates rollouts consisting of queries (questions with multiple choice's options) and LLM responses using the current policy $\pi_\theta^{policy}$. These rollouts are distributed to all training groups for preference evaluation.

**Distributed Reward Generation:** Each group $g_i$ first uses the PluraLLM GPO Srewa et al. (2025) model to generate preference probabilities for the received rollouts. These probabilities are then converted to rewards $r_{g_i}$ in different ways. In our evaluation, we focus on two approaches: (1) preference

probability prediction where rewards are calculated directly from the predicted probabilities, while for (2) preference ranking, the probabilities are first converted to rankings before reward calculation. These task-specific rewards $\{r_{g_1}, r_{g_2}, \ldots, r_{g_l}\}$ reflecting how well the generated responses align with each group's preferences are transmitted back to the aggregation server.

**Adaptively Alpha Aggregation:**   The core of the federated RLHF framework is the aggregation of local models. At each training round, the central server receives rewards updates from different groups and combines them to form a new global reward model. We evaluated three standard aggregation methods and proposed a novel adaptive scheme.

We have $\mathbf{r} = \{r_{g_1}, r_{g_2}, r_{g_3}, \ldots, r_{g_l}\}$ from $N$ clients, where $r_i$ represents how good the LLM outputs align with client $i$ preference. Recent work in the literature introduced an aggregation method, namely alpha aggregationPark et al. (2024) which achieves the consensus among heterogeneous feedback in RLHF. This is highlighted in Equation 1. This consensus reward aggregation is controlled by $\alpha$, $Agg_\alpha(\mathbf{r}) = max(\mathbf{r})$, when $\alpha = \infty$ and $Agg_\alpha(\mathbf{r}) = min(\mathbf{r})$, when $\alpha = -\infty$.

$$Agg_\alpha(\mathbf{r}) = \begin{cases} \frac{1}{\alpha} \log(\frac{1}{N} \sum_{i \in N} \exp(\alpha r_i)), & \alpha \neq 0 \\ \frac{1}{N} \sum_{i \in N} r_i, & \alpha = 0 \end{cases} \tag{1}$$

We propose an adaptive aggregation technique that uses this alpha aggregation.

$$Agg_\alpha(\mathbf{r}) = \begin{cases} \frac{1}{|\mathcal{G}|} \sum_{g \in \mathcal{G}} r_{g_i} & \text{if } FI \approx 0.999 \\ \log\left(\frac{1}{|\mathcal{G}|} \sum_{g \in \mathcal{G}} \exp(\alpha_g^t \cdot r_{g_i})\right) & \text{otherwise} \end{cases} \tag{2}$$

To achieve a balanced accumulated alignment reward history $\mathbf{h} = \{h_{g_1}, h_{g_2}, h_{g_3}, \ldots, h_{g_l}\}$, across clients, the aggregation weights $\alpha_i$ are dynamically adjusted in inverse proportion to each client's historical alignment performance($\alpha_i = \texttt{softmax}(1 - h_i)$). Specifically, a client $i$ with a lower accumulated alignment reward, $h_i$, is assigned a higher weight, $\alpha_i$. As shown in Equation 2, a higher $\alpha_i$ value increases the dominance of the corresponding reward $r_i$. Hence, the $\alpha_i$ value for client $i$ changes adaptively based on the alignment history for this client $h_i$.

**PPO Training and Iteration:**   With the aggregated rewards $r^t$, the server performs PPO optimization to update the policy model $\pi_{\theta^t}^{policy} \to \pi_{\theta^{t+1}}^{policy}$. The updated policy generates new rollouts for the next iteration, and the process continues until a predefined number of iterations or specific alignment score is reached. This iterative approach ensures continuous adaptation to diverse group preferences while maintaining fairness through our adaptive aggregation scheme.

## 4 EVALUATION

We evaluate our approach using Gemma-2B-it, a fine-tuned version of the Gemma model, as our base LLM Team et al. (2024). Our experiments utilize the Pew Research Center's Global Attitudes Surveys dataset Durmus et al. (2023), which captures diverse public opinions across social, political, and economic issues from various demographic groups. More details on the experiment setup and configuration are summarized in Appendix A.

We assess performance using fairness metrics and alignment scores across two primary tasks: the *preference probability prediction task* (see Figure 3) and the *preference ranking task* (see Figure 4). Our evaluation framework encompasses various reward functions and aggregation strategies. We compare our adaptive alpha aggregation against standard federated approaches (Min, Max, Average) and a supervised fine-tuning (SFT) baseline.

### 4.1 EVALUATION FRAMEWORK

The LLM rollout $X$ consists of questions and responses generated by the policy model $\pi_\theta(q_i)$ for each question $q_i$. We parse these responses to extract the relevant outputs and denote the parsed LLM response as $y_i^{\text{llm}}$ for question $i$. This parsed response is then used to compute rewards across different metrics.

Let $q_i$ denote a question, $o_{i,j}$ the $j$-th option for question $i$, and $y_i^{\text{llm}}$ the parsed LLM response for question $i$. Each group generates rewards $r_{g_i}$ by comparing the LLM output $y_i^{\text{llm}}$ against their preference data (PluralLLM predictions Srewa et al. (2025)) $p^{\text{GPO}}$.

## 4.2 REWARD METRICS

Our evaluation employs two categories of reward metrics to assess alignment quality across different aspects of preference modeling.

### 4.2.1 DISTANCE-BASED REWARD METRICS (PREFERENCE PREDICTION TASK)

These rewards capture alignment between LLM predicted and GPO target probability distributions:

**Wasserstein Reward:** Measures optimal transport cost between distributions.

$$r_{g_i}^{\text{Was}} = \frac{W_1(y_i^{\text{llm}}, p_i^{\text{GPO}})}{n-1} \tag{3}$$

$r_{g_i}^{\text{Was}} \in [0, +\infty]$, where 0 indicates perfect distribution match. Lower values indicate better alignment between LLM and group preferences.

**Cosine Similarity Reward:** Captures directional similarity between preference vectors.

$$r_{g_i}^{\text{Cos}} = \frac{y_i^{\text{llm}} \cdot p_i^{\text{GPO}}}{||y_i^{\text{llm}}|| \cdot ||p_i^{\text{GPO}}||} \tag{4}$$

$r_{g_i}^{\text{Cos}} \in [-1, 1]$, where 1 indicates identical direction, 0 indicates orthogonal, and -1 indicates opposite direction. Higher values indicate better preference alignment.

**KL Divergence Reward:** Measures information-theoretic alignment.

$$r_{g_i}^{\text{KL}} = D_{KL}(p_i^{\text{GPO}}||y_i^{\text{llm}}) = \sum_j p_{i,j}^{\text{GPO}} \log \frac{p_{i,j}^{\text{GPO}}}{y_{i,j}^{\text{llm}}} \tag{5}$$

$r_{g_i}^{\text{KL}} \in [0, \infty)$, where 0 indicates identical distributions, more positive = greater divergence. Smaller values indicate better alignment.

### 4.2.2 RANKING-BASED REWARD METRICS (PREFERENCE RANKING TASK)

These rewards evaluate preference ordering consistency:

**Kendall Tau Reward:** Measures rank correlation between LLM and GPO orderings.

$$r_{g_i}^{\text{Ken}} = \tau(\text{rank}(y_i^{\text{llm}}), \text{rank}(p_i^{\text{GPO}})) \tag{6}$$

$r_{g_i}^{\text{Ken}} \in [-1, 1]$, where 1 indicates perfect rank agreement, 0 indicates no correlation, and -1 indicates perfect disagreement. Higher values indicate better ranking alignment.

**Borda Reward:** Position-weighted scoring based on ranking accuracy.

$$r_{g_i}^{\text{Bor}} = \frac{\sum_{j=1}^{n}(n-j+1) \cdot \mathbb{I}[\text{rank}(y_i^{\text{llm}})_j = \text{rank}(p_i^{\text{GPO}})_j]}{n(n+1)/2} \tag{7}$$

$r_{g_i}^{\text{Bor}} \in [0, 1]$, where 1 indicates perfect position-wise ranking match, 0 indicates no correct positions. Higher values indicate better ranking quality.

**Binary Reward:** Simple correctness indicator.

$$r_{g_i}^{\text{Bin}} = \mathbb{I}[\text{rank}(y_i^{\text{llm}}) = \text{rank}(p_i^{\text{GPO}})] \tag{8}$$

$r_{g_i}^{\text{Bin}} \in \{0, 1\}$, where 1 indicates exact ranking match, 0 indicates any disagreement. Binary indicator of perfect alignment.

## 4.3 AGGREGATION SCHEMES

Let $\mathcal{G} = \{g_1, g_2, \ldots, g_l\}$ be the set of training groups. For each question $q_i$, the server aggregates the rewards $\{r_{g_1}, r_{g_2}, \ldots, r_{g_l}\}$ across groups using different strategies:

**Average Aggregation:**

$$r_{\text{final}}^t = \frac{1}{|\mathcal{G}|} \sum_{g \in \mathcal{G}} r_{g_i} \tag{9}$$

Provides balanced representation but may mask group-specific needs.

**Min Aggregation:**

$$r_{\text{final}}^t = \min_{g \in \mathcal{G}} r_{g_i} \tag{10}$$

Ensures no group is left behind but may be overly conservative, limiting overall performance.

**Max Aggregation:**

$$r_{\text{final}}^t = \max_{g \in \mathcal{G}} r_{g_i} \tag{11}$$

Optimizes for best-case performance but may neglect underrepresented groups.

**Adaptive Alpha Aggregation:**

$$r_{\text{final}}^t = \begin{cases} \frac{1}{|\mathcal{G}|} \sum_{g \in \mathcal{G}} r_{g_i} & \text{if } FI \approx 0.999 \\ \log \left( \frac{1}{|\mathcal{G}|} \sum_{g \in \mathcal{G}} \exp(\alpha_g^t \cdot r_{g_i}) \right) & \text{otherwise} \end{cases} \tag{12}$$

Dynamically balances fairness and performance by favoring historically underperforming groups.

The adaptive weights $\alpha_g^t$ are computed using reversed softmax on historical alignment scores:

$$\alpha_g^t = \frac{\exp((1 - h_g^{t-1})/T)}{\sum_{g' \in \mathcal{G}} \exp((1 - h_{g'}^{t-1})/T)} \tag{13}$$

with temperature $T = 0.1$ and $h_g^{t-1}$ being group $g$'s historical alignment score.

We provide an analysis and theoretical proof for convergence with the adaptive alpha aggregation in Appendix C.

### 4.4 FAIRNESS EVALUATION METRICS

The Fairness Index (FI) measures reward variation across groups for the same question-response pair:

$$FI = \frac{1}{|\mathcal{Q}|} \sum_{q_i \in \mathcal{Q}} \frac{1}{1 + \text{CoV}^2(q_i)} \tag{14}$$

where the Coefficient of Variation for question $q_i$ is:

$$\text{CoV}(q_i) = \frac{\sigma(\{r_{g,i}\}_{g \in \mathcal{G}})}{\mu(\{r_{g,i}\}_{g \in \mathcal{G}})} \tag{15}$$

$FI \in [0, 1]$, where 1 = perfect fairness (identical rewards across groups), 0 = maximum unfairness. Higher $FI$ values indicate more equitable treatment across demographic groups, while lower values suggest systematic bias favoring certain groups over others.

### 4.5 PREFERENCE PROBABILITY PREDICTION TASK RESULTS AND ANALYSIS

The SFT baseline demonstrates suboptimal performance with fairness indices ranging from 0.83-0.98 and consistently lower alignment scores, highlighting the need for preference-based alignment. Detailed quantitative results are shown in Table 1 across multiple reward functions in the Value task.

**Aggregation Strategy Comparison:** Our adaptive alpha aggregation consistently achieves superior fairness performance, reaching 0.99 across Wasserstein and Cosine metrics while maintaining competitive alignment scores. Notably, alpha aggregation demonstrates remarkable consistency, achieving the highest fairness indices across most reward types while preserving balanced average and minimum alignment scores.

**Reward Function Analysis:** Distance-based rewards (Wasserstein, Cosine, KL) substantially outperform ranking-based approaches (Kendall, Borda, Binary). Wasserstein and Cosine rewards with alpha aggregation achieve optimal fairness (0.99) while maintaining strong average alignment scores (0.90-0.95). The minimum alignment scores, crucial for ensuring no group is left behind, remain competitive (0.89-0.94), demonstrating effective fairness-performance balance.

**Key Insights:** The adaptive alpha approach effectively addresses the fairness-performance trade-off by dynamically weighting groups based on historical alignment scores. While Max aggregation occasionally achieves higher average alignment scores, it compromises fairness indices and minimum group performance, potentially leaving underrepresented groups behind.

### 4.6 PREFERENCE RANKING TASK RESULTS AND ANALYSIS

The Order task evaluation focuses on ranking-based metrics (Kendall Tau, Borda, Binary). More details are shown in Table 1. The SFT baseline, trained on population-averaged preferences, shows particularly poor performance in ranking tasks with fairness indices of 0.83-0.89 and substantially lower alignment scores (0.31-0.50 average, 0.25-0.41 minimum). This performance degradation in

Table 1: Fairness evaluation of pluralistic alignment across tasks, rewards, and aggregation strategies. FI = Fairness Index. Alignment scores are reported under multiple metrics (higher is better unless noted). Both average (Avg AS) and minimum (Min AS) alignment scores are shown. Highest values in each column are shown in **bold**, except for KL and Was columns where lowest values are highlighted.

| Task | Client Reward | Method | Server Agg. | Fairness Index (FI) | | | | | | Avg Alignment Score (Avg AS) | | | | | | Min Alignment Score (Min AS) | | | | | |
|---|---|---|---|---|---|---|---|---|---|---|---|---|---|---|---|---|---|---|---|---|---|
| | | | | Was. | Cos. | KL | Ken. | Bor. | Bin. | Was. | Cos. | KL | Ken. | Bor. | Bin. | Was. | Cos. | KL | Ken. | Bor. | Bin. |
| | — | SFT | — | 0.98 | 0.97 | 0.88 | 0.85 | 0.83 | 0.97 | 0.10 | 0.82 | 0.4 | 0.28 | 0.38 | 0.23 | 0.08 | 0.77 | 0.55 | 0.13 | 0.34 | 0.23 |
| Preference Prediction Task | WassersteinReward | PPO | Alpha | **0.99** | **0.99** | **0.94** | **0.91** | **0.86** | **1.00** | 0.05 | 0.90 | 0.26 | 0.30 | 0.42 | 0.22 | **0.06** | **0.89** | 0.26 | 0.21 | 0.37 | 0.22 |
| | | | Min | 0.98 | **0.99** | 0.93 | 0.87 | 0.79 | 0.90 | 0.05 | **0.91** | 0.22 | **0.42** | 0.44 | 0.27 | **0.06** | 0.89 | **0.27** | 0.34 | 0.41 | **0.28** |
| | | | Avg | **0.99** | **0.99** | **0.94** | **0.91** | **0.86** | **1.00** | 0.05 | 0.90 | 0.26 | 0.30 | 0.42 | 0.22 | 0.06 | 0.89 | 0.26 | 0.21 | 0.37 | 0.22 |
| | | | Max | **0.99** | **0.99** | 0.90 | 0.88 | 0.80 | 0.85 | **0.03** | 0.91 | 0.23 | 0.45 | **0.51** | **0.31** | 0.07 | 0.89 | 0.27 | **0.43** | **0.47** | 0.31 |
| | CosineReward | PPO | Alpha | **0.99** | **0.99** | 0.89 | **0.88** | **0.89** | **0.91** | 0.05 | 0.92 | 0.21 | 0.28 | 0.42 | 0.21 | **0.06** | 0.90 | 0.27 | **0.19** | 0.32 | **0.19** |
| | | | Min | **0.99** | **0.99** | 0.90 | 0.88 | 0.89 | 0.80 | 0.05 | 0.92 | 0.22 | **0.34** | **0.45** | **0.28** | 0.06 | 0.90 | 0.28 | 0.21 | 0.34 | 0.22 |
| | | | Avg | **0.99** | **0.99** | 0.89 | 0.88 | 0.89 | 0.91 | 0.05 | 0.92 | 0.21 | 0.28 | 0.42 | 0.21 | 0.06 | 0.90 | 0.27 | **0.19** | 0.32 | **0.19** |
| | | | Max | **0.99** | **0.99** | **0.91** | 0.87 | 0.88 | 0.88 | 0.05 | **0.93** | **0.19** | 0.31 | 0.42 | 0.22 | 0.06 | **0.91** | **0.24** | 0.20 | **0.34** | **0.19** |
| | KLReward | PPO | Alpha | **0.99** | **0.99** | **0.92** | **0.92** | **0.90** | 0.78 | 0.06 | **0.92** | 0.19 | 0.40 | 0.50 | 0.29 | 0.07 | **0.90** | **0.24** | **0.18** | 0.38 | 0.22 |
| | | | Min | **0.99** | **0.99** | 0.91 | 0.90 | 0.89 | **0.84** | 0.06 | 0.91 | **0.17** | **0.43** | **0.51** | **0.33** | 0.07 | 0.89 | 0.22 | 0.33 | **0.43** | 0.31 |
| | | | Avg | **0.99** | **0.99** | 0.91 | 0.89 | 0.89 | 0.76 | 0.06 | 0.91 | 0.19 | 0.40 | 0.48 | 0.27 | 0.06 | 0.89 | 0.26 | 0.29 | 0.40 | 0.25 |
| | | | Max | **0.99** | **0.99** | 0.91 | 0.90 | 0.86 | 0.75 | **0.04** | 0.91 | 0.19 | 0.33 | 0.40 | 0.20 | **0.05** | 0.88 | **0.24** | 0.22 | 0.33 | **0.19** |
| | KendallTauReward | PPO | Alpha | **0.99** | **0.99** | **0.96** | 0.90 | 0.71 | **0.91** | 0.07 | 0.75 | 0.48 | 0.43 | 0.38 | **0.29** | 0.08 | 0.72 | **0.55** | 0.34 | 0.36 | **0.28** |
| | | | Min | **0.99** | **0.99** | 0.95 | 0.90 | 0.75 | 0.91 | 0.07 | 0.73 | **0.49** | **0.45** | **0.39** | 0.29 | 0.08 | 0.71 | 0.54 | 0.37 | 0.36 | 0.28 |
| | | | Avg | **0.99** | **0.99** | 0.94 | 0.90 | **0.76** | 0.91 | 0.07 | 0.74 | 0.48 | 0.45 | 0.39 | 0.29 | 0.08 | 0.71 | 0.54 | **0.38** | **0.37** | 0.28 |
| | | | Max | **0.99** | **0.99** | 0.94 | **0.92** | 0.73 | 0.91 | **0.06** | **0.76** | 0.44 | 0.44 | 0.38 | 0.28 | **0.07** | **0.73** | 0.50 | 0.37 | 0.35 | 0.28 |
| | BordaReward | PPO | Alpha | **0.99** | **0.99** | 0.96 | **0.89** | 0.71 | 0.91 | 0.08 | 0.73 | 0.50 | 0.43 | **0.39** | **0.29** | 0.09 | 0.69 | 0.57 | 0.35 | **0.36** | 0.28 |
| | | | Min | **0.99** | **0.99** | **0.98** | 0.86 | 0.69 | **0.92** | 0.09 | 0.73 | **0.52** | **0.44** | 0.39 | 0.28 | 0.10 | **0.70** | 0.58 | **0.37** | 0.36 | 0.28 |
| | | | Avg | **0.99** | **0.99** | 0.97 | 0.89 | 0.71 | 0.91 | 0.09 | 0.73 | 0.51 | 0.42 | 0.38 | 0.29 | 0.10 | 0.70 | 0.58 | 0.35 | 0.36 | 0.28 |
| | | | Max | **0.99** | **0.99** | 0.97 | 0.89 | 0.71 | 0.90 | 0.08 | **0.74** | 0.49 | 0.44 | 0.39 | 0.29 | **0.09** | 0.70 | 0.56 | 0.36 | 0.36 | 0.28 |
| | BinaryReward | PPO | Alpha | **0.99** | **0.99** | 0.96 | **0.89** | **0.68** | **1.00** | 0.08 | **0.74** | 0.52 | **0.42** | 0.38 | 0.28 | **0.10** | **0.70** | 0.60 | 0.34 | 0.35 | **0.28** |
| | | | Min | **0.99** | **0.99** | **0.98** | 0.86 | 0.66 | **1.00** | 0.10 | 0.70 | **0.70** | 0.37 | 0.37 | 0.28 | 0.11 | 0.66 | **0.77** | 0.30 | 0.35 | 0.28 |
| | | | Avg | **0.99** | **0.99** | 0.97 | 0.89 | 0.67 | **1.00** | 0.09 | 0.73 | 0.54 | 0.42 | 0.38 | **0.29** | 0.10 | 0.70 | 0.59 | **0.35** | **0.36** | 0.28 |
| | | | Max | **0.99** | **0.99** | 0.97 | 0.89 | 0.68 | **1.00** | 0.08 | 0.73 | 0.56 | 0.41 | 0.38 | 0.28 | **0.10** | 0.70 | 0.64 | 0.33 | 0.34 | 0.28 |
| Preference Ranking Task | — | SFT | — | — | — | — | 0.89 | 0.87 | 0.83 | — | — | — | 0.38 | 0.50 | 0.31 | — | — | — | 0.25 | 0.41 | 0.27 |
| | KendallTauReward | PPO | Alpha | — | — | — | **0.92** | 0.81 | **0.97** | — | — | — | **0.58** | 0.47 | **0.36** | — | — | — | **0.47** | 0.42 | **0.31** |
| | | | Min | — | — | — | **0.92** | 0.81 | **0.99** | — | — | — | 0.52 | 0.46 | 0.30 | — | — | — | **0.43** | 0.41 | 0.28 |
| | | | Avg | — | — | — | **0.92** | 0.82 | 0.90 | — | — | — | 0.50 | 0.48 | 0.33 | — | — | — | 0.40 | 0.40 | 0.28 |
| | | | Max | — | — | — | 0.91 | **0.88** | 0.80 | — | — | — | 0.47 | **0.53** | 0.35 | — | — | — | 0.35 | **0.44** | 0.28 |
| | BordaReward | PPO | Alpha | — | — | — | **0.94** | **0.95** | 0.86 | — | — | — | **0.53** | **0.61** | **0.39** | — | — | — | 0.34 | **0.45** | 0.28 |
| | | | Min | — | — | — | **0.92** | 0.91 | **0.89** | — | — | — | 0.47 | 0.53 | 0.30 | — | — | — | 0.36 | 0.44 | 0.28 |
| | | | Avg | — | — | — | 0.93 | 0.92 | 0.86 | — | — | — | 0.49 | **0.58** | **0.39** | — | — | — | 0.35 | **0.47** | **0.31** |
| | | | Max | — | — | — | 0.91 | **0.92** | 0.78 | — | — | — | 0.45 | 0.54 | 0.32 | — | — | — | 0.34 | **0.45** | 0.28 |
| | BinaryReward | PPO | Alpha | — | — | — | **0.91** | 0.89 | 0.79 | — | — | — | **0.49** | 0.53 | **0.35** | — | — | — | 0.33 | 0.42 | 0.25 |
| | | | Min | — | — | — | 0.90 | 0.83 | **0.90** | — | — | — | 0.49 | 0.49 | 0.34 | — | — | — | **0.39** | 0.41 | **0.28** |
| | | | Avg | — | — | — | 0.91 | 0.90 | 0.79 | — | — | — | 0.49 | 0.53 | 0.35 | — | — | — | 0.37 | **0.44** | 0.28 |
| | | | Max | — | — | — | 0.91 | **0.91** | 0.79 | — | — | — | 0.47 | **0.54** | 0.35 | — | — | — | 0.35 | **0.44** | 0.28 |

ranking tasks underscores how averaged preference training fails to capture the nuanced ordering preferences that vary significantly across demographic groups.

**Ranking Reward Performance:** Alpha aggregation with Kendall Tau rewards achieves the highest fairness index (0.92) and superior average alignment scores (0.58) compared to the SFT baseline (0.38). Borda rewards demonstrate the strongest overall performance, reaching fairness indices up to 0.95 with alpha aggregation, and achieving the highest average alignment scores (0.61).

Figure 2 demonstrates that **ALPHA aggregation (blue circles) consistently provides the best fairness–performance trade-off,** occupying or approaching the upper-right quadrant (FI > 0.9, Min AS > 0.3) in all panels. For *KendallTauReward*, ALPHA attains high FI and strong worst-group performance across metrics (Kendall: FI≈0.92, Min AS≈0.47; Borda: 0.81, 0.42; Binary: 0.97, 0.31), whereas MIN pushes FI high on Binary (≈0.99) but *hurts* Min AS (≈0.28). For *BordaReward*, ALPHA achieves the *highest* Min AS across metrics (Kendall ≈0.53, Borda ≈0.61, Binary ≈0.39) with top/tied FI (Kendall ≈0.94, Borda ≈0.95, Binary ≈0.86). For *BinaryReward*, ALPHA again yields the largest Min AS (Kendall ≈ 0.49, Borda ≈ 0.53, Binary ≈ 0.35) with competitive FI, outperforming MIN/AVG/MAX in protecting the worst-served group. Across panels, the SFT baseline underperforms, reinforcing the need for federated preference alignment. Overall, the visualization shows that ALPHA most effectively resolves the fairness–performance tension by *maximizing worst-group (Min AS) performance at high FI.*

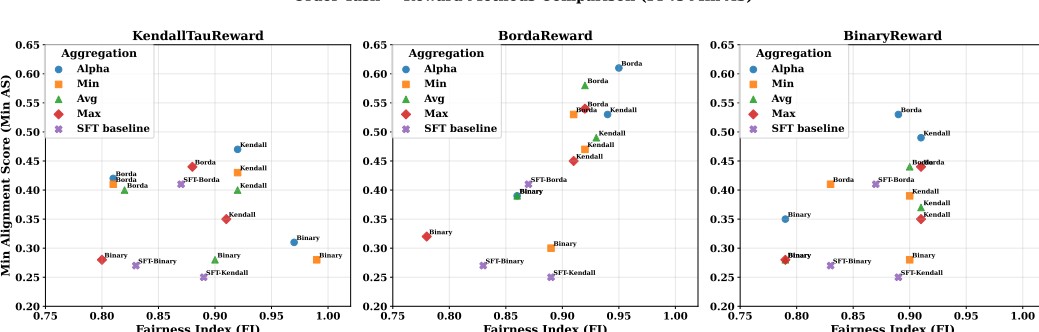

Figure 2: **Order task — FI vs. Min AS (worst–group performance) with reward-as-aggregation.** Each subplot fixes *one reward as the main aggregation metric* on the server— *KendallTauReward* (left), *BordaReward* (middle), and *BinaryReward* (right)— and then measures its effect on the three ranking metrics (Kendall, Borda, Binary). Points are server aggregation strategies (ALPHA, MIN, AVG, MAX); SFT baselines are purple crosses. We use the **minimum alignment score (Min AS)** on the y-axis because it reflects the performance of the *worst-served group*, while the x-axis shows the Fairness Index (FI).

**Aggregation Strategy Impact:** Across all ranking rewards, alpha aggregation maintains competitive performance while consistently achieving better fairness indices than alternative aggregation strategies. Importantly, our evaluation of minimum alignment scores reveals that alpha aggregation successfully prevents the marginalization of lowest-performing groups, maintaining minimum scores (0.31-0.47) that are competitive with or superior to other approaches, while simultaneously achieving higher average performance.

**Recommendations:** For order-based tasks, we recommend Borda rewards with alpha aggregation, which provides the optimal balance between fairness (0.94-0.95) and alignment performance (0.53-0.61 average). This combination effectively captures group ranking preferences while maintaining equitable treatment across demographic groups.

### 4.7 OVERALL ASSESSMENT

Our adaptive alpha aggregation demonstrates superior performance across both task types, consistently achieving the highest fairness indices while maintaining competitive alignment scores. The approach successfully addresses the critical challenge of preventing any group from being left behind, as evidenced by competitive minimum alignment scores across all evaluation scenarios. These results validate our hypothesis that adaptive weighting based on historical alignment performance provides an effective mechanism for achieving equitable federated learning in preference alignment tasks.

## 5 LIMITATIONS AND FUTURE WORK

While our study demonstrates the effectiveness of adaptive alpha aggregation for pluralistic alignment, several areas present natural directions for further research.

**Underlying RL Framework.** Our current implementation relies on PPO. While effective, PPO can be computationally expensive. Future work should explore more resource-efficient alternatives such as GRPO or DPO, which would allow testing the aggregation strategy across different optimization paradigms and at larger scales.

**Model and Dataset Scope.** We evaluate on Gemma-2B-it using the Pew Research Global Attitudes dataset, which may be relatively conducive to cross-group alignment. Broader validation on base models and domains with more adversarial or conflicting preferences would provide a stronger stress test of the method's robustness.

**Task Generalization.** Our experiments focus on multiple-choice Q&A tasks, which offer a controlled setting for evaluation. Extending the framework to diverse tasks such as summarization, dialogue, or code generation would demonstrate its wider applicability and highlight how aggregation impacts more open-ended alignment scenarios.

These considerations do not detract from our main contribution—a systematic evaluation framework with a novel adaptive aggregation scheme—but rather open exciting avenues for expanding its applicability across models, datasets, and tasks.

## 6 CONCLUSION

In conclusion, our work addresses the critical challenge of evaluating LLM alignment in decentralized, federated environments. We have demonstrated that the choice of aggregation technique is not a trivial detail but a fundamental evaluation protocol that directly shapes a model's fairness and performance. Our systematic evaluation of standard aggregation methods, alongside the introduction of an adaptive scheme, provides a clear framework for assessing the trade-offs between alignment and fairness. The results show that our proposed Adaptive Alpha Aggregation achieves a superior balance, offering a practical path toward developing truly pluralistic and equitably aligned LLMs. This research contributes a valuable evaluation methodology to the field and opens up new avenues for future work, including applying this framework to a broader range of tasks and model architectures.

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

## A    EXPERIMENT CONFIGURATIONS AND HYPERPARAMETERS

Table 2: SFT configuration and hyperparameters.

| Hyperparameter | Value |
|---|---|
| *Model* | |
| Base model | `google/gemma-2-2b-it` |
| Precision | BF16 |
| *Data / Task* | |
| Train/valid split | 80/20 |
| Max sequence length | 500 (include prompt and response) |
| *LoRA Adapter* | |
| Rank ($r$) | 16 |
| Alpha | 32 |
| Dropout | 0.05 |
| *Optimization* | |
| Batch size (per device) | 16 |
| Gradient accumulation steps | 4 |
| Learning rate | $5 \times 10^{-5}$ |
| Scheduler | cosine |
| Warmup steps | 150 |
| Weight decay | 0.01 |
| *Training* | |
| Epochs | 1 |

Our experimental setup begins with supervised fine-tuning (SFT) as outlined in Table 2. We use the Gemma-2-2b-it model as our base, employing LoRA adaptation with rank 16 for efficient parameter updates. The SFT training utilizes a cosine learning rate scheduler with warmup and is conducted for a single epoch to establish our baseline model.

As summarized in Table 3, both the policy and value models in PPO are initialized from the SFT model. During training, we employ two distinct prompt formats for evaluation: a preference probability prediction task requiring models to assign probability scores to all options, and a preference ranking task requiring complete ordinal ranking from most to least preferred (see Figures 3 and 4). Our implementation builds upon the Hugging Face TRL library von Werra et al. (2022). All experiments were conducted on 3 nodes, each equipped with A100 GPUs, Intel(R) Xeon(R) Gold 6326 CPUs @ 2.90GHz, and 256GB RAM.

## B    GROUP PREFERENCE OPTIMIZATION (GPO)

Group preference alignment refers to techniques designed to adapt LLM outputs to reflect the distinct preferences, values, or judgments of different groups or demographics. GPO Zhao et al. (2023) was introduced as a few-shot alignment framework that steers LLMs toward group-specific preferences. GPO augments the base LLM with an independent transformer module, trained via in-context supervised learning with only a handful of samples to predict group preferences and refine model outputs. This module acts as a preference model for different groups, learning distinct alignment patterns across diverse communities. By leveraging an in-context autoregressive transformer, GPO enables flexible and efficient alignment, allowing LLMs to adapt dynamically to varying user preferences. In PluralLLM, GPO is used as a preference predictor.

---

[1]Rewards are whitened over each rollout before PPO updates.

Table 3: PPO configuration and hyperparameters (policy and value models initialized from the SFT model).

| Hyperparameter | Value |
|---|---|
| *General* | |
| Policy model | Gemma 2 SFT model |
| Value model | Gemma 2 SFT model |
| *Model / Quantization* | |
| Quantization | 4-bit (`nf4`, double-quant = True) |
| Compute dtype | BF16 |
| Attention implementation | eager |
| *LoRA (PEFT)* | |
| Rank ($r$) | 32 |
| Alpha | 32 |
| Dropout | 0.05 |
| *Optimization* | |
| Per-device train batch size | 4 |
| Gradient accumulation steps | 24 |
| Learning rate | $1 \times 10^{-5}$ |
| Optimizer | AdamW |
| Weight decay | 0.0 |
| Scheduler | linear |
| *PPO Trainer* | |
| PPO epochs | 2 |
| Mini-batches | 8 |
| Per-device eval batch size | 32 |
| Response length | 42 |
| Temperature | 0.6 |
| KL coefficient | 0.05 |
| Clip range | 0.2 |
| Clip range (value) | 0.2 |
| Value loss coefficient ($v_f$) | 0.2 |
| Discount factor ($\gamma$) | 1.0 |
| GAE lambda ($\lambda$) | 0.95 |
| Reward whitening | Per rollout (before PPO update)[1] |

## C    THEORETICAL JUSTIFICATION AND CONVERGENCE ANALYSIS

This section provides a formal theoretical justification for the adaptive alpha formulation and analyzes its convergence properties. We demonstrate that our proposed algorithm, when applied to a non-convex optimization landscape, converges to a stationary point.

### C.1    PROBLEM FORMULATION

We consider the federated learning problem of aligning a global model $\theta$ with a set of $N$ user groups, where each group $g \in 1, \ldots, N$ has its own objective function $L_g(\theta)$. The goal is to minimize a weighted sum of these local objectives, where the weights are dynamically adjusted:

$$\min_\theta F(\theta) = \sum_{g=1}^{N} \alpha_g^t L_g(\theta) \tag{16}$$

At each iteration $t$, the adaptive weight $\alpha_g^t$ is a function of the historical alignment score $h_g^{t-1}$ of group $g$. We define the update rule for the global model as a federated gradient descent step:

$$\theta^{t+1} = \theta^t - \eta \nabla F(\theta^t) = \theta^t - \eta \sum_{g=1}^{N} \alpha_g^t \nabla L_g(\theta^t) \tag{17}$$

where $\eta$ is the learning rate.

---

**Preference Probability Prediction Prompt**

```
<bos><start_of_turn>user
```
You are an expert in modelling group preferences. You will receive **a question and exactly 4 options**.

**Your task**
- Assign a preference score to **each and every option**
- Produce **4** scores—no option may be skipped or combined
- Each score must be a decimal between 0 and 1, and **the rounded scores must sum to 1.00**
- Higher scores represent options a typical group is more likely to choose

**Output format**
- One line, comma-separated decimal numbers, **no spaces**
- Round each to **2** decimal places
- **No extra text, labels, or symbols**
- Example: 0.65,0.20,0.10,0.05

Return ONLY the 4 scores in the same order as options.

**Question:** Germany's influence in the EU Options: A: Has too much influence B: Has too little influence C: Has about the right amount of influence D: DK/Refused
```
<end_of_turn> <start_of_turn>model
```

Figure 3: Preference Probability Prediction Prompt

---

**Preference Ranking Prompt**

```
<bos><start_of_turn>user
```
You are an expert in ranking group preferences. You will receive **a question and exactly 4 options**.

**Your task**
- Rank all **4 provided options** from most to least preferred
- Process every option—no skipping or combining
- Order options based on what a typical group would most likely choose
- Higher preference options appear first

**Output format**
- One line, comma-separated option letters, **no spaces**
- Use the exact provided letters
- **No extra text, labels, or symbols**
- Example: B,C,A,D

Return ONLY the 4-letter ranking.

**Question:** Germany's influence in the EU Options: A: Has too much influence B: Has too little influence C: Has about the right amount of influence D: DK/Refused
```
<end_of_turn> <start_of_turn>model
```

Figure 4: Preference Ranking Prompt

## C.2 ASSUMPTIONS

To prove convergence, we make the following standard assumptions on the local objective functions $L_g(\theta)$:

**Assumption 1: Smoothness.** Each local loss function $L_g(\theta)$ is $L$-smooth, meaning that for any $\theta, \theta' \in \mathbb{R}^d$, we have:

$$L_g(\theta') \leq L_g(\theta) + \nabla L_g(\theta)^T(\theta' - \theta) + \frac{L}{2}\|\theta' - \theta\|^2 \tag{18}$$

This implies that the gradients are bounded.

**Assumption 2: Bounded Gradients.** The gradients of the local loss functions are bounded. There exists a constant $M$ such that for any group $g$ and iteration $t$:

$$\|\nabla L_g(\theta^t)\|^2 \leq M^2 \tag{19}$$

**Assumption 3: Bounded Historical Scores.** The historical alignment scores $h_g^t$ are bounded within a positive range, $0 < \epsilon \leq h_g^t \leq H$ for all $g,t$. This is a reasonable assumption given that alignment scores are typically normalized metrics.

### C.3 PROOF OF CONVERGENCE

Our proof relies on constructing a Lyapunov function and showing that it decreases with each iteration. We use the objective function value $F(\theta^t)$ as our Lyapunov function.

**Lemma 1.** Under Assumption 1, the objective function $F(\theta)$ is also $L$-smooth.

*Proof:*

$$F(\theta') = \sum_{g=1}^{N} \alpha_g^t L_g(\theta') \tag{20}$$

$$\leq \sum_{g=1}^{N} \alpha_g^t \left( L_g(\theta) + \nabla L_g(\theta)^T (\theta' - \theta) + \frac{L}{2} \|\theta' - \theta\|^2 \right) \tag{21}$$

$$\leq \sum_{g=1}^{N} \alpha_g^t L_g(\theta) + \left( \sum_{g=1}^{N} \alpha_g^t \nabla L_g(\theta) \right)^T (\theta' - \theta) + \frac{L}{2} \left( \sum_{g=1}^{N} \alpha_g^t \right) \|\theta' - \theta\|^2 \tag{22}$$

Since the weights are a softmax-like distribution, $\sum_{g=1}^{N} \alpha_g^t = 1$

$$F(\theta') \leq F(\theta) + \nabla F(\theta)^T (\theta' - \theta) + \frac{L}{2} \|\theta' - \theta\|^2 \tag{23}$$

Thus, $F(\theta)$ is $L$-smooth.

**Proposition 1.** *Let $\theta_t$ be the sequence of model parameters generated by the adaptive alpha federated learning algorithm. Under Assumptions 1-3, if the learning rate $\eta \leq \frac{1}{L}$, the algorithm converges to a stationary point, i.e., $\lim_{t \to \infty} \mathbb{E}[\|\nabla F(\theta_t)\|^2] = 0$.*

*Proof.* From the $L$-smoothness of $F(\theta)$, we have:

$$F(\theta^{t+1}) \leq F(\theta^t) + \nabla F(\theta^t)^T (\theta^{t+1} - \theta^t) + \frac{L}{2} \|\theta^{t+1} - \theta^t\|^2 \tag{24}$$

Substitute the update rule $\theta^{t+1} - \theta^t = -\eta \nabla F(\theta^t)$ :

$$F(\theta^{t+1}) \leq F(\theta^t) - \eta \|\nabla F(\theta^t)\|^2 + \frac{L}{2} \eta^2 \|\nabla F(\theta^t)\|^2 \tag{25}$$

$$F(\theta^{t+1}) \leq F(\theta^t) - \eta \left( 1 - \frac{L\eta}{2} \right) \|\nabla F(\theta^t)\|^2 \tag{26}$$

Rearranging the terms, we get:

$$\eta \left( 1 - \frac{L\eta}{2} \right) \|\nabla F(\theta^t)\|^2 \leq F(\theta^t) - F(\theta^{t+1}) \tag{27}$$

Summing over $T$ iterations:

$$\sum_{t=0}^{T-1} \eta \left( 1 - \frac{L\eta}{2} \right) \|\nabla F(\theta^t)\|^2 \leq F(\theta^0) - F(\theta^T) \tag{28}$$

Since $F(\theta)$ is bounded below (as loss functions are non-negative), $F(\theta^T) \geq 0$, so $F(\theta^0) - F(\theta^T)$ is bounded. $\sum_{t=0}^{T-1} \|\nabla F(\theta^t)\|^2 \leq \frac{F(\theta^0) - F(\theta^T)}{\eta(1 - \frac{L\eta}{2})}$ Dividing by $T$ and taking the limit as $T \to \infty$:

$$\lim_{T \to \infty} \frac{1}{T} \sum_{t=0}^{T-1} \|\nabla F(\theta^t)\|^2 \leq \lim_{T \to \infty} \frac{F(\theta^0) - F(\theta^T)}{T\eta(1 - \frac{L\eta}{2})} = 0 \tag{29}$$

This proves that the average squared norm of the gradient converges to zero, which implies that the algorithm converges to a stationary point. This holds regardless of the specific form of the weights $\alpha_g^t$, as long as they sum to 1. The adaptive nature of the weights ensures that the stationary point found is one that balances the individual group objectives, aligning with the core motivation of the adaptive alpha formulation. $\qquad\square$

C.4 ADAPTIVE ALPHA AS A DYNAMIC REGULARIZATION MECHANISM

The adaptive alpha formulation is not merely a static weighting scheme; it introduces a dynamic regularization mechanism that facilitates both exploration in divergent preference landscapes and convergence to a fair, robust solution. We formalize this property as a proposition.

**Proposition 2.** *The adaptive alpha formulation introduces a dynamic regularization term that perturbs the gradient direction towards groups with historically low alignment, thus promoting exploration and mitigating local minima that marginalize specific groups.*

*Proof Sketch.* The global gradient at iteration $t$ is given by $\nabla F(\theta^t) = \sum_{g=1}^{N} \alpha_g^t \nabla L_g(\theta^t)$. The adaptive alpha weight for a group $g$ is inversely proportional to its historical performance $h_g^{t-1}$.

$$\alpha_g^t = \frac{\exp(-\beta h_g^{t-1})}{\sum_{j=1}^{N} \exp(-\beta h_j^{t-1})} \tag{30}$$

where $\beta$ is a hyperparameter controlling the sensitivity of the adaptation.

Consider a scenario where the model has converged to a solution that is optimal for a majority of groups but performs poorly on a single group, say $k$. In this case, the historical alignment score $h_k^{t-1}$ for group $k$ would be significantly lower than the other groups' scores. Consequently, the adaptive weight $\alpha_k^t$ for this group will be disproportionately high.

The gradient update for the next step, $\nabla F(\theta^t)$, will be heavily influenced by $\nabla L_k(\theta^t)$. This effectively means the model is "pulled" in the direction that reduces the loss for the under-aligned group $k$, even if it slightly increases the loss for other groups. This dynamic adjustment acts as a regularization term, preventing the optimization from settling into a sub-optimal local minimum that ignores minority preferences.

We can view the global gradient as the sum of two components:

$$\nabla F(\theta^t) = \sum_{g \neq k} \alpha_g^t \nabla L_g(\theta^t) + \alpha_k^t \nabla L_k(\theta^t) \tag{31}$$

The second term, $\alpha_k^t \nabla L_k(\theta^t)$, serves as a dynamic regularization signal that guides the model away from an unfair solution. As the model improves its alignment with group $k$, its historical score $h_k^{t-1}$ will increase, and the weight $\alpha_k^t$ will decrease, effectively reducing the influence of this regularization term. This process repeats dynamically for all groups, ensuring a continuously fair and balanced convergence path. This mechanism is crucial for navigating non-convex landscapes where simple averaged gradients can lead to poor local minima. □

