# OpenReview forum: "A Systematic Evaluation of Preference Aggregation in Federated RLHF for Pluralistic Alignment of LLMs"
_ICLR.cc/2026/Conference — ICLR 2026 Conference Withdrawn Submission_

### Official Review · Reviewer_cS56 · 2025-10-31

**Soundness:** 2
**Presentation:** 2
**Contribution:** 3
**Rating:** 4
**Confidence:** 3

**Summary:**

To address the privacy concern and risk of biases toward a narrow demographic for RLHF, Federated Learning has been integrated with it. However, this presents a new challenge: How to aggregate the diverse and potentially conflict preference signals from different user groups? Different aggregation methods determine whose preferences are prioritized and whose are marginalized. To shed light on this problem, this paper proposes a evaluation framework to measure the impact of different aggregation techniques on both alignment performance and fairness. Besides, it also proposes an adaptive aggregation scheme to achieve better trade-off between alignment and fairness.

**Strengths:**

1. This paper concentrates on an important question of RLHF with federated learning, i.e., the method to aggregate diverse preference signals which are associated with the fairness.
2. A systematic evaluation method is proposed.
3. A new adaptative aggregation strategy is proposed.

**Weaknesses:**

1. Several main important modules need more clarification
- How RLHF with federated learning is conducted together with the evaluation system? How the parameters are updated?
- Details about the evaluation set Pew Research Center’s Global Attitudes Surveys dataset.
- Details about the preference prediction task and preference ranking task.
2. Experiments on more types of experiments are required to verify the practicality and generalizability of the whole evaluation framework, as the Pew Research Center’s Global Attitudes Surveys dataset maybe different from the popular QA-style in realistic interactions. Datasets in natural QA style maybe necessary and more valid.
3. These experimental results are very close, and a significance test may be required.
4. I have another concern about the motivation of RLHF with Federated Learning. RLHF itself only needs preference labels as the supervision signals but not the original log data, while RLHF with FL still has access to each group’s preference toward the rollouts. In this case, what privacy was really protected?

**Questions:**

1. The citation format is wrong.
2. How does the PluraLLM GPO method (in Line 106) work?

---

### Official Review · Reviewer_vwNN · 2025-10-31

**Soundness:** 2
**Presentation:** 1
**Contribution:** 2
**Rating:** 2
**Confidence:** 3

**Summary:**

This paper explores integrating RLHF with federated learning (FL) to aggregate diverse, potentially conflicting preference signals from different user groups. Specifically, it proposes a systematic evaluation framework to analyze the impact of various preference aggregation techniques. This work trains Gemma-2B-it with various client reward methods and server aggregation approaches, and evaluates the trained models across three metrics: fairness index, average alignment score, and minimum alignment score. Empirical studies show that alpha aggregation achieves the best performance.

**Strengths:**

1. This paper measures the trained performance across numerous metrics, providing a convincing and comprehensive evaluation.
2. The paper considers quite a number of client reward methods and server aggregation approaches and shows their performance under different combinations. I can see that the authors put a great deal of effort into the experiments.

**Weaknesses:**

1. I find that the paper is weak in surveying the related work. Some papers, such as FedBiscuit [1], are supposed to be discussed and compared in the experiments.
2. I am pretty confused by this work. In Section 3, the authors state that they train the policy model $\pi_{\theta}^{policy}$ using PPO. As I know, the PPO under RLHF requires a reward model and a policy model. However, I cannot find the reward model. Instead, the work aggregates rewards but does not explain how they are obtained. Authors should make this point clearer, including whether the rewards are calculated using the same set of queries.
3. The authors should discuss the objective of the LLM (policy model). Is the model used to predict the distribution of options across groups of individuals? After the policy model is trained, will it be distributed back to each group again?
4. The work heavily relies on a previous algorithm named PluraLLM. However, the authors do not provide brief details in the manuscript. As a reader, I find it really hard to follow this work without the background. The main text of the paper is just slightly more than seven pages, which is still far away from the page limit (i.e., nine pages). The authors should make these details available in the paper.
5. In the experiments, the authors train the work with Gemma-2B-it and draw a conclusion that adaptive alpha aggregation is the best aggregation approach. I wonder whether the conclusion still holds when applied to other base models, such as gemma-2-2b or gemma-3-4b.
6. This paper may use \cite command in the LaTeX rather than \citet, making all citations look weird, e.g., "as our base LLM Team et al. (2024)." The authors should correct the weird citation, i.e., "as our base LLM (Team et al., 2024)."

**Reference:**
[1] Towards Federated RLHF with Aggregated Client Preference for LLMs, ICLR 2025

**Questions:**

**See weaknesses**

---

### Official Review · Reviewer_oqYr · 2025-11-01

**Soundness:** 2
**Presentation:** 2
**Contribution:** 2
**Rating:** 4
**Confidence:** 3

**Summary:**

This paper proposes an adaptive aggregation framework for federated RLHF that dynamically reweights group preferences based on historical alignment performance to improve fairness. It introduces a systematic evaluation protocol to analyze the trade-off between overall alignment quality and equitable representation across diverse user groups. Experiments on QA tasks show the method consistently enhances fairness without compromising alignment.

**Strengths:**

1.	This paper addresses a timely problem: fair preference aggregation in federated RLHF for pluralistic LLM alignment.
2.	Proposes a systematic evaluation framework with comprehensive experiments across reward types and aggregation schemes.
3.	The adaptive weighting strategy effectively boosts fairness without task demonstrations or demographic data.

**Weaknesses:**

1.	The adaptive alpha aggregation is a heuristic extension of existing work, offering limited technical novelty.
2.	Experiments are limited to multiple-choice QA with model-generated preferences; generalization to open-ended tasks or real human feedback remains unverified.
3.	Evaluation relies solely on Gemma-2B-it; results may not generalize to larger or architecturally different LLMs, limiting the robustness of conclusions.

**Questions:**

See the weaknesses.

---

### Official Review · Reviewer_YE4P · 2025-11-03

**Soundness:** 2
**Presentation:** 1
**Contribution:** 2
**Rating:** 2
**Confidence:** 2

**Summary:**

This paper claims to solve the problem of pluralistic alignment for in the federated learning setting. They adapt a reward aggregation strategy called alpha aggregation (originally introduced in Park et al. (2024)) and weigh the clients such that higher weight is given to the less aligned client's data. There is some experiment on evaluating different aggregation equations using Gemma-2B-it.

**Strengths:**

The problem is well motivated, as LLMs are deployed widely, we will be gathering private data from users and considering the problem of pluralistic data with private data in the federated learning setting is quite interesting.

**Weaknesses:**

1. My most major concern is regarding the readability of the paper. It is not clear what the user types are: does each client contain a different user? or are groups of clients assigned to the same user types? It seems like the variables l and N are used interchangeably? It also took me a while to understand what the evaluation metrics mean. Significant work needs to be put in to make the paper more readable.
2. Experiments with just a 2B is too small and at least a 7B model experiments are the norm for alignment.
3. If the main contribution is in using the alpha-aggregation for reward aggregation the contribution is a bit weak in terms of novelty.

**Questions:**

Is " Average 0and" a typo in the abstract? Also, I am very confused with the notations as mentioned in the weakness section.

---

### Note · Authors · 2025-11-14

I have read and agree with the venue's withdrawal policy on behalf of myself and my co-authors.